# Multimodal Deep Learning for Predicting Adverse Birth Outcomes Based on Early Labour Data

**DOI:** 10.3390/bioengineering10060730

**Published:** 2023-06-19

**Authors:** Daniel Asfaw, Ivan Jordanov, Lawrence Impey, Ana Namburete, Raymond Lee, Antoniya Georgieva

**Affiliations:** 1School of Computing, University of Portsmouth, Portsmouth PO1 3HE, UK; 2Nuffield Department of Women’s & Reproductive Health, University of Oxford, Oxford OX1 2JD, UKantoniya.georgieva@wrh.ox.ac.uk (A.G.); 3Department of Computer Science, University of Oxford, Oxford OX1 3QG, UK; 4Faculty of Technology, University of Portsmouth, Portsmouth PO1 2UP, UK

**Keywords:** CTG, FHR, deep learning, CNN, LSTM

## Abstract

Cardiotocography (CTG) is a widely used technique to monitor fetal heart rate (FHR) during labour and assess the health of the baby. However, visual interpretation of CTG signals is subjective and prone to error. Automated methods that mimic clinical guidelines have been developed, but they failed to improve detection of abnormal traces. This study aims to classify CTGs with and without severe compromise at birth using routinely collected CTGs from 51,449 births at term from the first 20 min of FHR recordings. Three 1D-CNN and LSTM based architectures are compared. We also transform the FHR signal into 2D images using time-frequency representation with a spectrogram and scalogram analysis, and subsequently, the 2D images are analysed using a 2D-CNNs. In the proposed multi-modal architecture, the 2D-CNN and the 1D-CNN-LSTM are connected in parallel. The models are evaluated in terms of partial area under the curve (PAUC) between 0–10% false-positive rate; and sensitivity at 95% specificity. The 1D-CNN-LSTM parallel architecture outperformed the other models, achieving a PAUC of 0.20 and sensitivity of 20% at 95% specificity. Our future work will focus on improving the classification performance by employing a larger dataset, analysing longer FHR traces, and incorporating clinical risk factors.

## 1. Introduction

Cardiotocography (CTG) is a continuous and simultaneous measurement of fetal heart rate (FHR) and maternal uterine contraction signals. CTG is commonly performed during or preceding labour to assess fetal wellbeing and reduce its mortality and morbidity [1]. Interpretation of the CTG patterns requires assessing the FHR baseline, variability, accelerations, and decelerations by a trained clinician. However, due to the complexity of CTG signals, visual interpretation is often challenging and imprecise [2], leading to miss diagnoses [3,4]. In the United Kingdom (UK), each year between 2015 to 2018, on average 125 intrapartum still births, 154 neonatal deaths, and 854 severe injuries were registered [5]. These adverse outcomes frequently lead to litigation. In England, in 2020/2021, over £4.1 billion was spent on settling obstetric claims, 59% of which were clinical negligence payments [6]. Enhancing the accuracy of CTG interpretation has the potential to enable clinicians to intervene earlier, thereby potentially preventing some of these adverse outcomes. This, in turn, can alleviate the substantial financial burden on the healthcare system. Globally in 2019, an estimated 2 million babies were stillborn [7]. Most these adverse outcomes that occurred during intrapartum periods are potentially preventable with CTG monitoring and appropriate interventions.

CTG remains at the center of the decision-making process in intrapartum fetal monitoring despite its limitations, as there is no other technology or method that has been shown to be as effective in assessing fetal well-being during labour. CTG is typically performed at maternity admission units/triage wards or after admission to the labour ward and in some countries like Sweden it is routinely performed, while in others, such as the UK, CTG is not recommended for low-risk births (about 40% of all) [8,9]. Under some circumstances, the initial 20–30 min CTG recording is called ‘admission CTG’ [10], and its role is controversial: some studies show that it may increase the incidence of unnecessary caesarean sections, especially in ‘low risk’ pregnancies [10], while others report its benefit in the decision to perform caesarean delivery when administered routinely to all births [8]. This lack of consensus can be attributed to the imprecision of current clinical guidelines and the poor sensitivity and specificity of the available tools to interpret CTG patterns. During the evaluation of 27,000 high-risk births, our team noted significant differences in the first-hour CTG features (extracted using objective computerized methods) and clinical risk factors between births with severe compromise and those without severe compromise [11].

Several automated methods have been proposed to address the subjective visual interpretation of CTG recordings [12,13,14,15]. Research efforts have been devoted to developing techniques that can automatically detect characteristics of the CTG signal [16,17,18,19]. These studies focus on detecting or quantifying abnormal patterns by mimicking what clinical rules and guidelines suggest or what experts do during their visual assessment. Such methods are commercially available but have not shown clinical benefit in randomised clinical trials [20,21] and have not been widely adopted. Other studies have used advanced signal processing techniques to extract multiple features from the CTG signal alone or in conjunction with clinical risk factors and apply machine learning approaches, including hierarchical Dirichlet process mixture models [22], logistic regression [23], neural networks [24], support vector machines [14,25], random forests [15], Bayesian classifier [26], and XGBoost [15,27], to find patterns from the extracted features. One of the key issues with the conventional machine learning models is that they require careful design of feature extractors that transform the input CTG signal into compact representations or feature vectors [28].

More recent studies on computerised CTG analysis apply modern deep learning (DL) techniques [12,29,30,31]. Convolutional Neural Networks (CNN) are among the most notable DL approaches that learn automatically abstract hierarchical representations directly from the input data using multiple hidden layers [28]. CNN models have been extensively applied in various medical data analysis tasks involving image and time series analysis [32,33]. Long Short-Term Memory (LSTM) networks are another class of DL models suitable for analysing sequential (time series) data [34]. Most previous works incorporating deep learning methods for the CTG analysis have used primarily 1D-CNN based networks. For example, Comert et al. [12], and Baghel et al. [31] analysed the last 90 min CTG data of the CTU-UHB open-access dataset [35], using 1D CNNs. Zhao et al. [12] reported better performance on the same dataset when implementing 2D CNNs by firstly transforming the CTG signal into 2D images using recurrence plots. More recently, Liu et al. [36] proposed attention based CNN-LSTM network with features of discrete wavelet transformation to analyse the CTU-UHB dataset. However, these studies have been evaluated mostly on small datasets (40–160 abnormal birth outcomes), which are prone to large within-class variability and between-class similarity, resulting in a model with poor generalisation performance. Furthermore, although the proposed deep learning models are valid, due to absence of a distinct holdout (testing) sets in the CTU-UHB dataset and the less rigorous evaluation approach adopted (using the whole dataset for the cross validation instead of only on the training set), the results should be interpreted with caution. On a much larger dataset, Petrozziello et al. [37] and Mohannad et al. [38] evaluated the performance of CNNs to classify CTGs. Petrozziello et al. [37] achieved promising performance using multimodal 1D CNNs to predict cord acidemia from the last hour CTG signal. Mohannad et al. [38] applied a multi-input CNN network to analyse CTG plots of the initial 30 min of the last 50 min before delivery and gestational age to predict foetuses with a low Apgar score. Overall, previous studies on analysis of CTG using deep learning approaches focused on the last hour recordings mostly using small datasets.

In this study, we present three deep learning models for prediction of birth outcome using FHR traces recorded around the onset of labour in both: the time domain, implementing a combination of 1D CNNs and LSTMs; and in the frequency domain, employing a 2D CNNs [33,39]. The models are trained to classify new-borns with and without severe compromise at birth. To our knowledge, this study represents a pioneering effort in the application of deep learning techniques for analyzing CTG traces during early labour. Given the absence of published results from computer-based methods, we conducted a comparison of our findings with the existing standards of clinical care. We hypothesise that DL methods trained with a large clinical dataset of CTGs from around the onset of labour could hold the potential to ultimately assist clinicians in identifying foetuses who are already compromised or are vulnerable at labour onset and may thus be at high risk for further injury during labour.

## 2. Materials and Methods

### 2.1. Data and Pre-Processing

#### 2.1.1. Description of the Dataset

This was a retrospective cohort study of infants delivered at the John Radcliffe Hospital in Oxford, UK, using a clinical data collection system between 1993 and 2012. The study received ethical approval from the Newcastle & North Tyneside 1 Research Ethics Committee, Reference 11/NE0044 (data before 2008), and from the South Central Ethics Committee, Reference 13/SC/0153 (for data after 2008). Informed consent by the participants was not required.

The clinical protocol has been to administer intrapartum CTG only to pregnancies deemed at ‘high-risk’. From them, 51,449 CTG tracings include births at gestation ≥36 weeks, longer than 1 h, and have no second stage trace in the first hour (Figure 1). In these records, 452 are births with a severe compromise—a composite outcome of intrapartum stillbirth, neonatal death, neonatal encephalopathy, seizures, and resuscitation followed by over 48 h in the neonatal intensive care unit. The rest of the cohort samples are labelled as no severe compromise.

#### 2.1.2. Pre-Processing

The CTG datasets are obtained with standard fetal monitors at a 4 Hz sampling rate for the FHR and 2 Hz for the uterine contraction signals. Consistently with our prior work [37], the FHR signals are down sampled to 0.25 Hz. We apply a two-stage pre-processing procedure to deal with the noise and missingness: signal cleaning; and gap imputation.

#### 2.1.3. Signal Cleaning

A bespoke algorithm is applied to remove artefacts from the CTG signal, for example, erroneous maternal heart rate capture and extreme outliers (FHR measurements >230 or <50 beats per minute (bpm)). The start time of the signal is adjusted to ensure adequate signal quality: the CTG is analysed using a sliding 5-min window (with a one-minute stride) to ensure that the signal loss is less than 50% in the first five minutes. The extracted cleaner 20-min FHR tracing has less than 50% signal loss, i.e., any sample with signal loss greater than the threshold is discarded. The cleaning process reduced the signal loss of the no severe compromise group (mean, ±std) from 26.9% (26.7, 27.2%) to 12.3% (12.2, 12.4%); and for the severe cases from 29.1% (26.6%, 31.5%) to 13.4% (12.3, 14.5%).

#### 2.1.4. Gap Imputation

Signal noise and loss are common in CTG tracings, resulting in both short (few seconds) and long (many minutes) gaps in the signal [40]. Following efficient noise removal, reliable gap imputation is an important task in data analysis and pre-processing phases, which is expected to improve the classifier performance at the later learning stage. There are a number of techniques proposed for inferring and imputing the gaps in FHR signals recorded by CTG, including Linear interpolation [37], Cubic spline interpolation [41], Sparse representation with dictionaries [42], Gaussian processes (GP), and others [43,44]. We compared the performance of the Linear, GP, and Autoregressive (AR) imputation techniques (example shown in Figure 2) based on their effect on the performance of the CNN, which is evaluated on the testing set. Since the AR consistently outperformed the Linear and GP gap imputation techniques, and achieved the highest accuracy the CNN’s classification accuracy, we used it to impute the gaps in the FHR signals of our dataset (results of the comparison between the gap imputation techniques is reported in [45]).

#### 2.1.5. Transformation of the FHR Signal to a 2D Image

The raw FHR signals (each sample 20 min long) are transformed into time-frequency using Fourier and Wavelet transforms, and the resulting images are analysed using the well-established 2D-CNNs. The spectrogram of Short-Time Fourier Transform (STFT) represents the normalised, squared magnitude of short-time Fourier transform coefficients [46]. To convert the input 1D FHR signal using spectrograms, the time domain signals are divided into shorter segments (windows), and Fourier transform is computed for each segment to obtain the frequencies. We used a 1 Hz FHR signal in the spectrogram and scalogram (Wavelet transform) analysis since it produced better accuracy in our preliminary experiment. The spectrogram is the STFT of each short signal segment, computed by sliding the window with a constant stride and an overlap through the entire record. In this work, we investigate the effect of different window strides and overlapping sizes on the classifier’s performance. The FHR signals are converted into spectrogram images by applying STFT given with (1).
(1)$$X(n,\omega )=\sum_{m=0}^{L-1}x\left[m\right]w[m-n]e^{-j2\Pi m\omega /L}$$
where x[m] is the input FHR signal, w[m] is the window, and *L* is the window length. X(n,ω) STFT of a windowed data centred at time point n and the log values of X(n,ω) are represented as spectrogram (128 × 128) images. Since the window length *L* is a hyperparameter, we investigated the effect of different window sizes on the classification performance. In our initial experiments the 128 × 128 spectrogram images lead to better classifier performance than 64 × 64 or 256 × 256 image sizes. Thus, we used the 128 × 128 image size for the rest of our analysis. 

In addition to the STFT spectrograms, we also investigated the wavelet scalograms. A scalogram is a time-frequency representation with a wavelet basis instead of sinusoidal functions. Like the Fourier spectrogram, the scalogram analyses compute the coefficients using sliding windows called wavelets, i.e., the input signal is multiplied with the wavelet at different time locations. The process is repeated by increasing the scale of the wavelet (also known as the mother wavelet). This dilation and contraction operation captures long- and short-time events from the input, where the dilated wavelet is sensitive to long-time events, and the contracted wavelet to short-time events [47]. The wavelet transform of a signal *x*(*t*) is defined as the integration of the *x*(*t*) with the shifted or scaled shapes from a mother wavelet ${ѱ}_{a,b}(t)$ as shown in (2).
(2)$${WT}_{x}(a,b)=\frac{1}{\sqrt{a}}\int_{-\infty }^{\infty }x(t)ѱ(\frac{t-b}{a})dt$$
where *a* is a scale parameter, *b* is a translation parameter, and ѱ(t) is the mother wavelet function. By using different scale factors of the wavelet transform, ${WT}_{x}$ computes wavelet coefficients of the signal at different scales. The absolute values of these continuous coefficients define the scalogram (in our case, represented as 128 × 128 image). The choice of the mother wavelet is important as the time-frequency analysis represents the match between the wavelet and the FHR signal. Here, we compare the Gaussian of order 8 (‘gaus’), Morlet (‘morl’), Shannon (‘shan’), and Mexican Hat (‘mexh’) wavelets. [47].

### 2.2. Deep Learning Models

#### 2.2.1. Data Augmentation

The dataset used in this study is substantially imbalanced: there are fewer positive samples (*n* = 384) than the negative ones (*n* = 43,293). The class imbalance in the dataset can make the learning process very challenging for any classifier and usually leads to poor prediction performance [48]. In our initial model training, we observed signs of overfitting. To mitigate this problem, we augmented the data of the severe compromise class, which we expected to act as a regulariser and improve the generalisation performance of the classifier. Several data augmentation methods have been proposed for time series signals, such as flipping the signal, adding noise, masking the segment of the signal [49]. We developed a simple data augmentation approach, tailored to our specific task, which involved extracting additional 20-min FHR segments from the first 1-h FHR data with 50% overlap, thereby increasing the size of positive samples by a factor of 4. Only the 20-min segments with less than 50% signal loss were augmented. The positive instances are then further oversampled by a factor of 2, which led to their overall increase by a factor of 8 in the training dataset. In CTG analysis, the under-sampling of the negative samples is a common practice [12,30], but in our experiments, these techniques did not improve the generalisation performance of the trained models.

#### 2.2.2. Deep Learning Architectures

The proposed architectures primarily constitute 1D-CNN and LSTMs. Three variations are investigated:(i)5-layer 1D-CNN network, in which the encoder is composed of five 1D-CNN layers and two fully connected (FC) layers;(ii)CNN-LSTM sequential architecture, in which the network has 5-layer 1D-CNN followed by 2-layer LSTM component and two FC layers;(iii)5-layer CNN-LSTM parallel architecture, in which a 5-layer 1D-CNN and two-layer LSTM networks are connected in parallel, followed by 2 FC layers.

The architectures of the three models are shown in Figure 3. We also employed a 2D-CNN of the FHR signal to analyse the spectrograms and scalograms. The network architecture comprises of 2D-CNNs (with ReLU activation), along with five residual blocks, followed by an average pooling layer. The residual blocks are a 2D-CNN with skip connections, a widely used network architecture for various image recognition tasks (28). Each residual block in the skip connection has a 2D-CNN component, followed by a sequence of batch normalisation, dropout, and max pooling operations (Figure 4). In addition, we investigated the effect of different kernel sizes on classification performance as well.

The combined 1D-CNN-LSTM and 2D-CNN deep learning network architecture is a two-channel network: one channel to analyse the input 1D-FHR signal using 1D-CNN-LSTM parallel topology, while the other channel uses a 2D-CNN with skip connections (Figure 4) to extract spectral features from the sample. This makes the architecture capable of analysing and capturing the signal’s temporal and spectral characteristics. The input FHR signal (1D signal) and the corresponding spectrogram or scalogram (2D) are simultaneously fed to the respective channels. Subsequently, the output of the two channels is concatenated and fed to an FC layer.

The input to each of the 1D-CNN-LSTM based models is prepared as (B, T, F), where B, T and F represent the batch size, the length of times slices, and the signal dimension, respectively. In a 20-min FHR signal sampled at 0.25 Hz, there are 300 time steps (15 per minute, T = 300, F = 1) in each CTG sample. Similarly, the input to the 2D-CNN network is arranged as (B, H, W, C), where B is the number of samples or the batch size, while W (128), H (128) and C (3) are the width, height and number of channels of the image respectively. A Sigmoid activation function is used at the last layer of all the networks to obtain the class probability prediction of each sample.

#### 2.2.3. Training Procedure

We split the data randomly into training (85%) and test (15%) sets, while preserving the class ratio in each subset. Ten-fold cross-validation was performed using the training dataset, in which 90% of the samples are used for training the model and the remaining 10% for validation. During each fold, the model is trained for a maximum of 400 epochs with early stopping to mitigate overfitting (with a window size of 50 epochs) by monitoring the partial area under the receiver characteristic curve (PAUC). After the optimal model parameters are obtained, the PAUC is evaluated on the test set. We report the average performance of the ten models evaluated on the test set.

Since our class distribution is unbalanced, weighted binary cross-entropy loss is used for training, where the weight is based on the inverse class frequency, i.e., during training, the loss function penalises more (by factor of 14) the misclassification of severe compromised cases. As mentioned above, the network is trained for 400 epochs, using Adam optimiser with an initial learning rate of 0.001, decayed by a factor of 2 every 50 epochs. The batch size is set to 128 and batch normalisation is used with default parameters as recommended in [50]. Furthermore, we used dropout with a probability of 0.3 in all layers and early stopping based on the PAUC. The tuned hyperparameter values, including the CNN module’s filter size, the number of filters used in each CNN layer, and the unit number of LSTM modules, are summarised in Table 1. The model is implemented using TensorFlow on an NVIDIA GTX 2080 Ti 12GB GPU machine.

To ensure that every feature in the data has the same level of importance, features are standardized using the z-score:$$z_{i}=(x_{i}-µ)/\sigma$$
where $x_{i}$ is the original value of sample i in the dataset; $z_{i}$ is the normalized value, µ is the mean, and σ is the standard deviation.

## 3. Results

### 3.1. Performance Metrics

The performance of the models is evaluated using the partial area under the receiver operating characteristic curve (PAUC) and the true positive rate (TPR = TP/(TP + FN)) at a 5% false-positive rate (FPR). The PAUC is the AUC between 0 and 10% false-positive rates (FPR = FP/(FP + TN)). The metrics are selected to assess the accuracy of the model only at a very low specificity, i.e., to detect adverse birth outcomes as accurately as possible, while minimising the rate of false positives. It is crucial to minimise unnecessary interventions, particularly early in the labour. ROC curve is a commonly accepted graphical plot that shows the performance of a binary classifier for all classification thresholds. It is based on TPR and FPR values, which are calculated from true positive (TP), false positive (FP), true negative (TN), and false-negative (FN) values. The AUC measures the area underneath the entire ROC curve from (0, 0) to (1, 1). We also considered the Precision, Recall and F1-score values of the three 1D-CNN and LSTM based models.

### 3.2. Hyperparameter Tuning

Optimal model hyperparameters are tuned using the Bayesian optimisation (BO) and Hyperband (HB), (BOHB) optimiser [51]. The selected values for the batch size, the number of filters in CNN each layer, the kernel size, the number of layers, and the optimisation functions are all summarised in Table 1.

### 3.3. Performance of the Proposed Models

The performance of the three 1D-CNN and LSTM based models is shown in Figure 5. The 1D-CNN-LSTM parallel architecture achieved higher Sensitivity and PAUC on the testing set than the 1D-CNN-LSTM sequential architecture. The difference between PAUC values of the 1D-CNN-LSTM parallel and 1D-CNN-LSTM sequential models are statistically significant (Mann-Whitney U tests, two-tailed, U = 10, *p* = 0.002). All other differences between the three models in terms of PAUC and Sensitivity at 0.95 specificity values are not statistically significant. The comparison of the three models in terms of Precision, Recall, and F1-score is shown in Table 2. The best performing 1D-CNN and LSTM based models (from the 10 models trained using a 10-fold cross-validation) are shown in Figure 6.

Table 3 shows the performance of the 2D-CNN and the multimodal architecture (combined 1D-CNN-LSTM and 2D-CNN). The 2D-CNN using scalogram analysis produced slightly better results compared to the case of using the spectrograms. However, the performance of the 2D-CNN (alone) and the multimodal architecture was inferior compared to the 1D-CNN-LSTM parallel architecture. This indicates that using the raw FHR (temporal representation) as input can lead to a better classification performance than the case relying on the time-frequency representations.

The performance metrics when varying the window size of the spectrograms and the kernel sizes of the 2D CNN model are given in Table 4. Three different kernel sizes are considered for the comparison: 3 × 3; 5 × 5; and 7 × 7. The rest of the hyperparameters, such as, batch size, number of layers, and number of filters are selected using cross-validation. The highest PAUC and the sensitivity performance at a Specificity of 0.95 is achieved using a window size (overlapping size) of 64 (32) with a 3 × 3 kernel. This indicates that the two classes can be better separated when the input FHR signals’ frequency content is analysed using a window size of about 1 min. Nevertheless, the performance of the spectrogram analysis is inferior compared to the 1D-CNN and LSTM based models.

Table 5 demonstrates the performance metrics and the influence of the different kernel sizes on the scalograms generated using a variety of wavelet functions. The proposed 2D-CNN achieved the highest classification results using a 3 × 3 kernel on scalograms generated with Mexican hat wavelet functions. This indicates a greater similarity between the input FHR signal and the Mexican hat, making it a preferable wavelet. The scalograms analysis showed slightly better separation between the two classes than the spectrograms. However, the wavelets’ performances were inferior to those of the 1D-CNN and LSTM based models.

In the combined 1D and 2D architectures, the 2D CNN models based on spectrograms and scalograms were aggregated with the best performing 1D model (the 1D-CNN-LSTM parallel architecture). In this analysis, the spectrograms were computed with Fourier transform using a window size of 64, while the scalograms were obtained implementing the Mexican hat wavelets. The scalograms achieved slightly better results than the spectrograms, but the combined model’s overall performance was inferior to the 1D-CNN-LSTM parallel construct, indicating that the time-frequency representation provides not enough useful content in separating the two classes. Conversely, the temporal features extracted using the 1D-CNN-LSTM parallel architecture separated better the two classes than when the features were extracted from the time-frequency representation using the 2D-CNN architecture.

### 3.4. Comparison with Clinical Practice and OxSys

We compared the 1D-CNN-LSTM parallel model to *OxSys 1.5* (3) and clinical benchmark (11). *OxSys* uses two FHR features and two clinical risk factors to analyze the entire FHR trace with a 15-min sliding window. While in clinical practice, clinicians consider not only the findings of CTG interpretation but also consider clinical risk factors when making the diagnosis. The TPR of detecting severe adverse outcomes by the *Clinical Practice, OxSys 1.5*, and our model is presented in Table 6. The TPR in clinical practice reported in [3], which is defined as the number of emergency deliveries, is based on a clinical decision for “presumed fetal compromise” as a proportion of the total number of babies with compromise (5/162) within 2-h of start of CTG recording. The FPR is the number of emergency deliveries, based on a clinical decision for “presumed fetal compromise”, where there was no compromise as a proportion of the total number of normal cases (108/27,652). The results show that the *OxSys 1.5* which is based on the entire CTG and clinical risk factors achieved highest sensitivity. 

We also compared the sensitivity between our 1D CNN and LSTM-based models and clinical practice, focusing on a similar FPR value of 0.4%. The results, presented in Figure 7, indicate that the sensitivity of the 1D CNN-LSTM parallel model falls slightly below the optimal sensitivity achieved in clinical practice (2.4% vs. 3.1%). However, it is important to interpret these findings with caution, as the results *of Clinical Practice* are based on clinical risk factors and the initial 2-h CTG recording, whereas our analysis is solely based on the initial 20 min of FHR recording.

### 3.5. Effect of the Pre-Processing on the Model’s Output

We investigated the relationship between the model’s output and the location and/or magnitude of signal loss within CTG segments. Table 7 shows Spearman’s correlation coefficient between the model predictions (range between 0 and 1) and the percentage of signal loss, number of gaps in the signal, the longest gap length, and its location. The signal loss summary statistics are computed before the gap imputation. The results show weak correlation between the model’s outputs and the signal loss summary statistics, indicating that our model predictions are largely independent of the magnitude and the location of signal loss.

### 3.6. Post-Hoc Analysis of the Models’ Prediction

We utilized concept attribution as a technique to enhance the explainability of our deep learning model [52]. Concept attribution seeks to identify the key features or concepts in the input data that exert the greatest influence on the model’s decision-making. In this case, we investigated whether the predictions generated by the deep learning model were related to the clinical features of FHR used for evaluating initial FHR traces. In the initial stages of labour, the FHR baseline and variability play a crucial role in the interpretation of CTG (11). Table 8 shows the relationship between the predictions of the model (1D-CNN-LSTM parallel architecture) and the standard clinical features of the FHR, such as FHR baseline and FHR short-term variability (STV). The testing set samples are divided into three groups, based on the quartiles of the predicted values from the 1D-CNN-LSTM network: low (≤25 percentile), medium (>25 and <75 percentile), and high (≥75 percentile). The samples with clinically low STV (STV ≤ 3 ms) are more likely to have high DL predictions signifying increased risk (39.1% vs. 12.1%), which is in line with current clinical understanding that diminished STV is a significant risk for fetal compromise. On the other hand, the model’s predictions do not appear to be associated with the FHR baseline.

## 4. Discussion

This study investigates the potential of implementing different deep learning architectures for predicting births with and without severe compromise outcomes, using the first 20 min of the FHR signals of more than 51,000 births, recorded as per clinical practice in a UK hospital during 1993–2012. From the designed and proposed DL architectures, the 1D-CNN-LSTM parallel topology model achieved superior classification performance compared to the other two developed models: 2D-CNN; and the multimodal architectures (combined 1D-CNN-LSTM and 2D-CNN). The suboptimal performance of the 2D-CNNs could be attributed to the lack of informative features in the time-frequency representations of the FHR signal. The post-hoc analysis also indicates that the performance of the 1D-CNN-LSTM model is not biased to signal loss, and its predictions are related to a degree to low STV values, which aligns with the clinical expectation of what is important in the initial FHR [11].

The sensitivity of CTG, based on initial hour in detecting severely compromised births, is not well established and there is limited evidence available. Lovers et al. [11] reported that the sensitivity of admission CTG is approximately 3.1% at 0.4% FPR (Figure 7). Our best model, the parallel 1D-CNN-LSTM model, achieved a slightly reduced sensitivity of 2.4% at a 0.4% FPR. This outcome is promising, particularly considering that clinical sensitivity in practice relies on evaluating the initial 2-h CTG data and incorporates various clinical risk factors. Nonetheless, the findings imply that our model has the potential to serve as a valuable aid for clinicians in identifying fetal distress and averting adverse birth outcomes.

Previous studies have also explored the potential of data-driven approaches in detecting abnormalities in CTG traces, focusing on the last hour CTG recording. For instance, Petrozziello et al., [37], demonstrated that a 1D CNNs model that employs more than 35,000 CTGs of the last hour recording can achieve higher TPR in predicting birth acidemia (pH < 7.05) than the clinical diagnosis (53% vs. 31% at about 15% FPR). Other studies (12, 30, 31), using a much smaller dataset and pH < 7.15 as an abnormal outcome, also implemented 1D CNNs to classify FHR signals. However, these works have focused on detecting birth acidemia based on the last hour CTG recording. When similar outcome groups are investigated (as in this work: with and without severe compromise), the OxSys 1.5 (3), achieved slightly higher TPR (43% at 14% FPR) than the clinical diagnosis (35% at 16% FPR) and our 1D-CNN-LSTM model (35% at 16% FPR) on a dataset of more than 22,000 CTGs. Nevertheless, this relatively higher accuracy results from analysing the entire FHR trace (in our case, it is based on the first 20 min only).

The main contributions of our work are: proposing and implementing DL models, based on uniquely large and detailed dataset allowing their successful training, validation, and testing; the clinically relevant definition of a rare severe compromise; and the focus on the first 20 min of the FHR, seeking an early warning for those fetuses that are unlikely to sustain the stress of labour due to pre-existing vulnerability. We capped the false positive rate to 5% and achieved sensitivity of 20%—an encouraging result given that most infants who would sustain severe compromise, are expected to do so later in labour. Also, given the fact that the false positive rate cannot be precisely defined, due to the routine nature of our data, which includes high rates of clinical intervention and censuring the data. The achieved performance, as compared to the clinical benchmark (as shown in Figure 7), is highly encouraging. This is particularly noteworthy because predicting adverse outcomes in clinical practice relies not only on CTG patterns but also on various risk factors such as abnormal fetal growth, antepartum hemorrhage, prolonged rupture of membranes, and meconium staining of the amniotic fluid [9]. Consequently, a model that provides an objective assessment of CTG, without imposing a significant computational burden (with trace prediction in under a second with the ready trained model), can serve as an integral part of a clinical decision support tool. By doing so, it could contribute to optimizing the allocation of clinical resources, allowing clinicians to focus on other crucial responsibilities. Finally, in our future work, we expect to further improve the model accuracy by incorporating clinical risk factors into the analysis.

Some limitations of our approach are also worth noting. While data augmentation has demonstrated some effectiveness in addressing label imbalance and reducing model overfitting, it is important to consider the potential risk of amplifying label noise within the dataset. Thus, it is necessary to acknowledge that data augmentation alone may not offer a complete solution to the challenges associated with learning from imbalanced datasets. This limitation is evident in the variance of the ten cross-validated models, as depicted in Figure 5. Future work should address this by employing other techniques, perhaps creating synthetic data using generative adversarial networks [53]. The classification performance should also be improved by analysing longer traces and incorporating uterine contraction signals and clinical risk factors (such as fetal gestation and maternal co-morbidities) into the model. Finally, our approach does not explain which segment of the input leads to a particular prediction. Therefore, future work will consider applying an attention layer to provide better explainability [54].

## 5. Conclusions

We developed and evaluated three different deep neural network architectures to classify a 20-min FHR segment, recorded around the onset of labour, to investigate their potential in providing very early warning and triage women in labour into high or low-risk groups for further monitoring and/or review. We achieved superior performance using the proposed 1D-CNN-LSTM parallel architecture: the best model achieved a sensitivity of 20% at 95% specificity. The results are clinically encouraging, considering the fact that the majority of the compromised babies are not expected to have demonstrated problems, and if any, they are challenging to detect at the onset of labour. It is important to note that there is also room to improve the model classification performance by analysing the entire FHR trace and incorporating clinical risk factors.

Although the proposed DL architecture achieved encouraging results on the holdout test set, it could also be tested on an external dataset to further validate its generalisation performance. In addition, the investigated models do lack explainability, and future work will incorporate an attention mechanism that will introduce it.

## Figures and Tables

**Figure 1 bioengineering-10-00730-f001:**
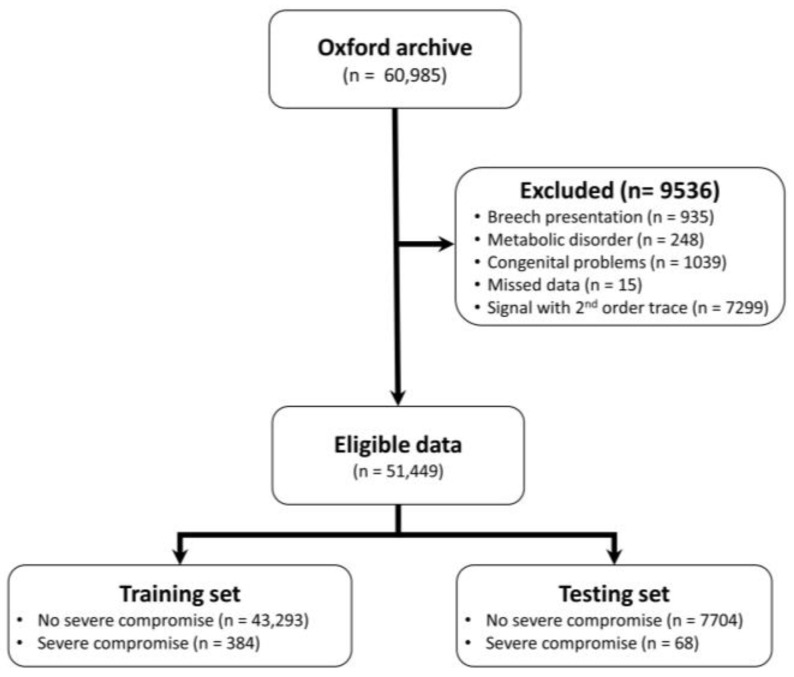
Data flow chart.

**Figure 2 bioengineering-10-00730-f002:**
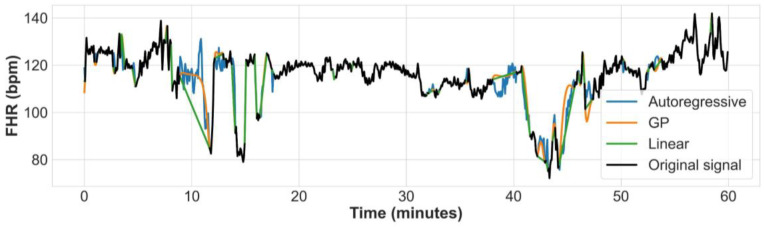
Example of an FHR signal from the dataset where gaps are imputed using Autoregressive, GP based, and Linear interpolation methods.

**Figure 3 bioengineering-10-00730-f003:**
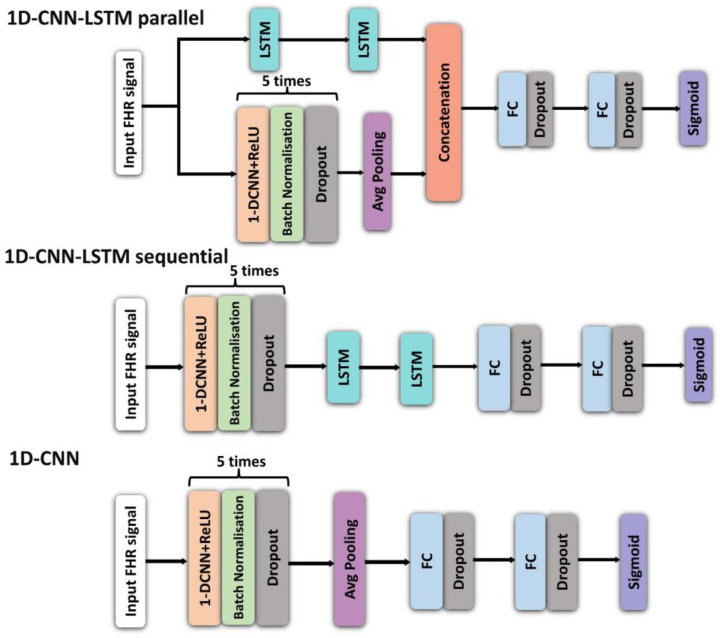
Architectures of: 1D-CNN-LSTM parallel (**top**); 1D-CNN-LSTM sequential (**middle**); and 1D-CNN (**bottom**) models (hyperparameters of these models are provided in Table 1).

**Figure 4 bioengineering-10-00730-f004:**
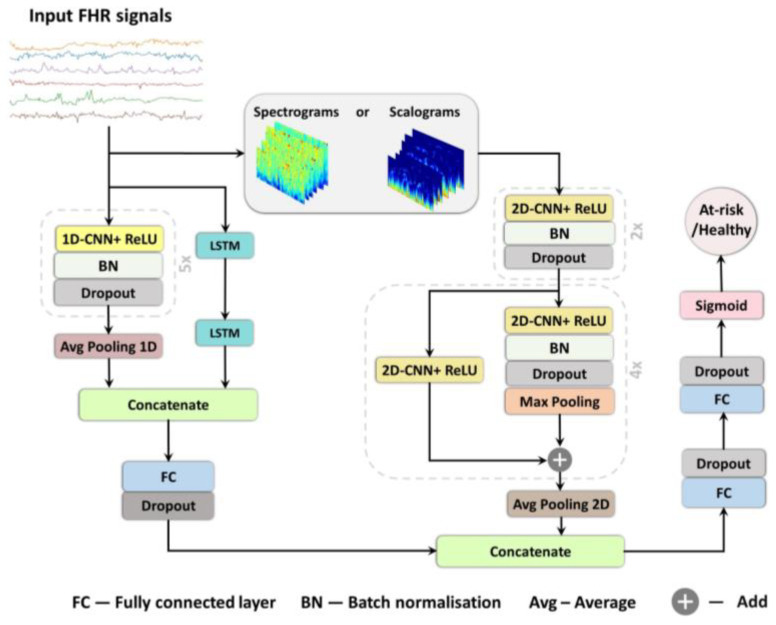
The proposed combination of 1D-CNN-LSTM and 2D-CNN architectures, used for the FHR signal classification.

**Figure 5 bioengineering-10-00730-f005:**
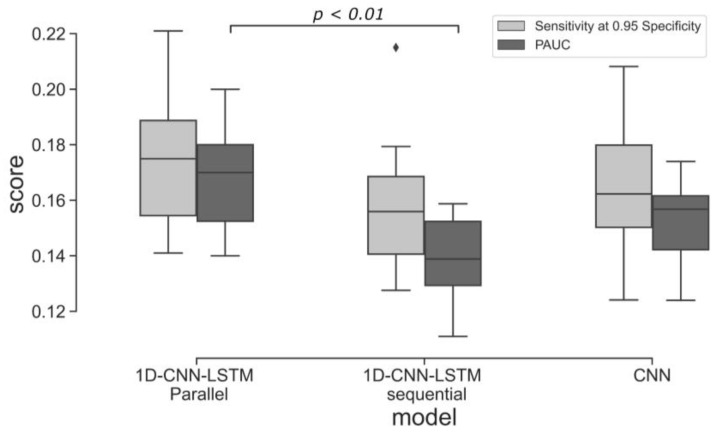
Classification performance (the ten models trained using 10-fold cross-validation evaluated on the test set) of the three 1D-CNN and LSTM based architectures.

**Figure 6 bioengineering-10-00730-f006:**
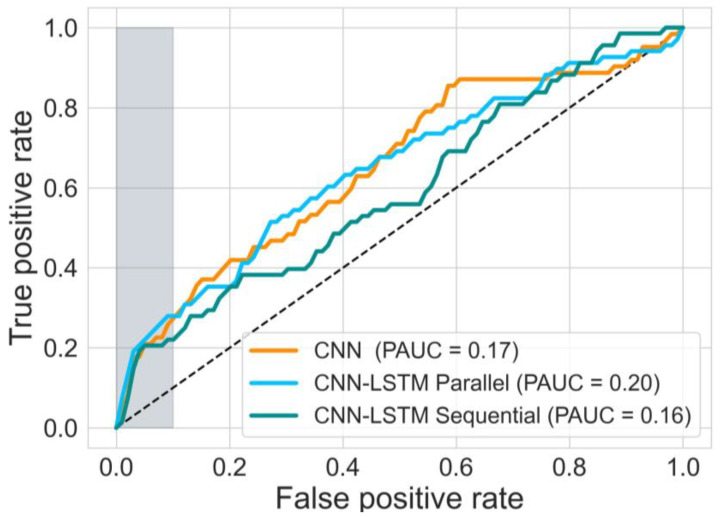
ROC of the best performing 1D-models from the 10-fold cross-validation, evaluated on the test set (68 severe compromises and 7704 samples without severe compromise).

**Figure 7 bioengineering-10-00730-f007:**
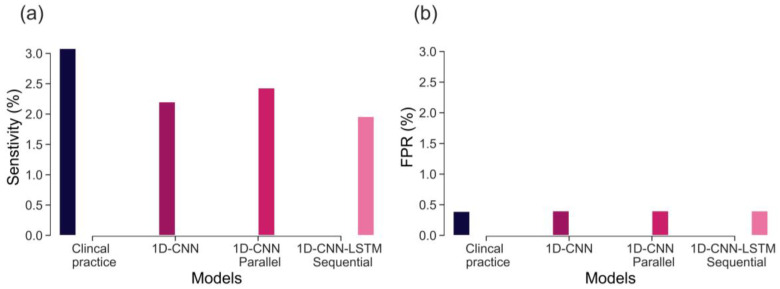
Performance comparison between 1D-CNN,1D-CNN-LSTM parallel, 1D-CNN-LSTM sequential, OxSys 1.5, and *Clinical Practice* (emergency deliveries in clinical practice for fetal distress cases within two hours of admission to labour ward) in terms of: (**a**) Sensitivity (TPR); and (**b**) FPR.

**Table 1 bioengineering-10-00730-t001:** Optimal model hyperparameters tuned for the parallel 1D-CNN-LSTM architecture. Note that the number of filters is for each of the five layers respectively.

Hyperparameter	Range	Optimal Value
Batch size	16, 32, 64, 128, 256, 512, 1024	512
Number of CNN layers	4, 5, 6, 7, 8	5
kernel size	3, 5, 7, 10, 15	3
Number of CNN filters	8, 16, 32, 64, 128	16, 32, 64, 64, 16
Optimization functions	SGD, RMSprop, Adam, Nadam	Adam
LSTM units	8, 16, 32, 64	16

**Table 2 bioengineering-10-00730-t002:** Classification performance (mean of the 10-fold cross-validation on the test set) of the 1D CNN and LSTM based architectures.

Model	AUC	Precision	Recall	F1-Score
1D-CNN	0.66	0.014	0.65	0.029
1D-CNN-LSTM sequential	0.61	0.012	0.49	0.025
1D-CNN-LSTM parallel	0.68	0.017	0.60	0.034

**Table 3 bioengineering-10-00730-t003:** Classification performance (mean of the 10-fold cross-validation on the test set) of the 2D CNN alone and when combined with 1D CNN-LSTM parallel architectures.

Method		PAUC	Sensitivity at 0.95 Specificity	AUC	Precision	Recall	F1-Score
Spectrogram	Only 2D CNN	0.12	0.13	0.59	0.010	0.48	0.019
	Combined with 1D-CNN-LSTM	0.15	0.14	0.60	0.011	0.45	0.021
Scalogram	Only 2D CNN	0.13	0.14	0.60	0.013	0.54	0.025
	Combined with 1D-CNN-LSTM	0.16	0.15	0.61	0.017	0.59	0.033

**Table 4 bioengineering-10-00730-t004:** Impact of the window length on the classification performance of the spectrograms. The results are performances of the best models from the 10-fold cross-validation, evaluated on the test set (the best classification performances are given in bold).

Window Length (Overlap Size)	Kernel Size	PAUC	Sensitivity at 0.95 Specificity
32 (16)	3 × 3/5 × 5/7 × 7	0.09/0.09/0.09	0.09/0.10/0.12
64 (32)	3 × 3/5 × 5/7 × 7	**0.11**/0.07/0.09	**0.13**/0.12/0.13
128 (64)	3 × 3/5 × 5/7 × 7	0.11/0.11/0.09	0.10/0.12/0.10
256 (128)	3 × 3/5 × 5/7 × 7	0.09/0.09/0.08	0.10/0.13/0.13

**Table 5 bioengineering-10-00730-t005:** Classification performance of the different wavelet functions and kernels sizes. The results are performances of the best models from the 10-fold cross-validation, evaluated on the test set (the best classification performances are given in bold).

Wavelet Function	Kernel Size	PAUC	Sensitivity at 0.95 Specificity
*Mexican hat* wavelet (mexh)	3 × 3/5 × 5/7 × 7	**0.13**/0.11/0.10	**0.14**/0.09/0.09
*Morlet* wavelet (morl)	3 × 3/5 × 5/7 × 7	0.11/0.11/0.11	0.12/0.12/0.11
*Shannon* wavelet (shan)	3 × 3/5 × 5/7 × 7	0.12/0.09/0.08	0.14/0.12/0.13
*Gaussian* wavelet (gaus8)	3 × 3/5 × 5/7 × 7	0.11/0.08/0.09	0.14/0.09/0.09

**Table 6 bioengineering-10-00730-t006:** Performance comparison between 1D-CNN-LSTM model, OxSys 1.5, and emergency deliveries in clinical practice for fetal distress cases.

Method	Sample Size		TPR	FPR
	Severe Compromise	No Severe Compromise		
OxSys 1.5 (entire CTG)	187	22,603	43.32%	16.45%
Clinical practice (first 2-h of CTG)	167	27,927	3.08%	0.39%
1D-CNN-LSTM (first 20 min)	68	7703	35.29%	16.16%

**Table 7 bioengineering-10-00730-t007:** Spearman correlation between the 1D-CNN-LSTM network predictions and signal loss (testing set data).

Data (n = 7772)	Signal Loss	Number of Gaps	Longest Gap Length	Location of Longest Gap
Severe compromise	0.10	0.05	0.12	0.07
No severe compromise	0.14	0.09	0.13	0.07

**Table 8 bioengineering-10-00730-t008:** Relationship between the model’s predicted outputs, the FHR STV and Baseline values (Spearman’s correlation coefficient (number of samples from the testing set)).

DL Predicted Values	STV ≤ 3	STV > 3	Baseline ≥ 150	Baseline < 150
Low	0.12 (99)	0.27 (1796)	0.24 (1613)	0.36 (282)
Medium	0.49 (401)	0.50 (3360)	0.53 (3564)	0.25 (195)
High	0.39 (321)	0.23 (1561)	1580)	0.39 (301)

## Data Availability

The data that support the findings of this study are available from the corresponding authors upon reasonable request.

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
