# Peer review of "Multimodal Deep Learning for Predicting Adverse Birth Outcomes Based on Early Labour Data"

_bioengineering, 2023, doi:10.3390/bioengineering10060730_

Round 1

Reviewer 1 Report

The manuscript presents a deep learning pipeline to address the problem of classifying and predicting severe compromise at birth from limited duration of CTG recordings (20 min). 
Despite the work is well written and address an interesting topic, there are some major points that need to be clarified and better discussed.

- The authors propose a deep learning approach on a very imbalanced dataset  Despite they declare the limitation of the study and attempt to address the imbalance issue by means of data augmentation, I would recommend clarifying and detailing more the data augmentation approach also explaining and discussing its reliability and potential risk of bias.

- The proposed models seem to be of limited performance, also when compared to clinical standards. The authors should better discuss and motivate the potential contribution and impact of the proposed pipeline in the clinical practice. How can this model impact the daily workflow? What are the computational efforts/timing? How the proposed approach translate into practice?

- No discussion nor results are provided regarding the explainability of the model, which is fundamental for clinical translation of AI based methodology. I would highly recommend attempting to provide clinically meaningful explanations of the model perfomances in order to discuss and assess the reliability of the proposed models.

- Are ethical approval or consent needed to use the data?

- Clinical translation is very limited. The authors should better explain and clarify why they consider their results promising from a clinical perspective (as they state in the Conclusion section).

- Are time-domain and frequency-domain features extracted with deep learning consistent with traditional CTG features?

- Please check and revise the caption of Figure 8

Author Response

Dear Reviewer,

We would like to express our gratitude for dedicating your valuable time to review our manuscript. Your insightful comments and suggestions are highly appreciated. We have thoroughly examined your feedback and have incorporated the necessary revisions into the manuscript. Please find the revised version attached in the accompanying file.

Thank you once again for your invaluable input.

Best regards,
Daniel (on behalf of all the authors)

Reviewer 2 Report

Dear authors,

it was a pleasure to read your paper. Intrapartum cardiotocography in everyday practice has crucial significance in the diagnosis of intrapartum fetal asphyxia and, interpretation of CTG tracing many times lead to unnecessary intervention, but in come cases fetal asphyxia remains unrecognized. As interpretation of CTG tracing is subjective method, misinterpretation is very likely. Therefore, objectification of the method may be useful. Your paper presenting deep learning model in CTG interpretation provides potential assistance in everyday work. You commented that this model should be further investigated. 

Introduction and methodology are presented correctly. results are presented textually, but tables 3 and 2 have been switched; table 7 is missing.

Discussion is well written, conclusion also.

However, references must be revised as there are references in which only first author is cited.

Author Response

(The authors gave the same response as above.)

Reviewer 3 Report

The authors developed a neural network model using CTG as input for monitoring fetal heart rate (FHR) during labor and assessing the health of the infant. This study compares different neural network architectures and the effect of different methods of extracting the time-frequency spectrum as input on the accuracy of the model.

1.       How are CTGs analyzed in clinical practice and what are the characteristics of CTGs with severe damage?

2.       What is the length of the input signal for wavelet, STFT?

3.       Where is Table 7?

4.       I am not quite able to understand the meaning of Table 8. What is the meaning of STV>3 and Baseline>=150? What I understand is that the authors want to prove that DL Prediction and STV<3 have a positive correlation, but what do the authors want to prove by putting the data in the other columns here?

5.       It is recommended to give the distribution of labels after data enhancement

6.       The split line of the table is missing.

7.       It is recommended to write the parameters used in the corresponding model structure in Figure 4.

8.       The layout of figures and tables is a bit messy, which makes the reading not smooth enough.

9.       Please bold the key information in the table, such as the highest value, and explain the meaning of the bold in the table title.

10.   The Window length and Kernel size of Table 4 and Table 5 should be given in the title.

11.   Table 2 and Table 3 use different evaluation metrics, and it is suggested to unify the evaluation metrics. It cannot be seen that the model accuracy will be improved after using 2D CNN.

12.   It is recommended to briefly describe the method of hyperparameter fine-tuning.

13.   Are there other relevant studies in the field? It is suggested to compare with the models used in these studies in the results section.

Good.

Author Response

(The authors gave the same response as above.)

Round 2

Reviewer 1 Report

The authors addressed all the concerns.
I would suggest further detailing the comparison with clinical standards.

Author Response

We would like to thank the reviewer for their prompt and thoughtful comments. In the revised manuscript, we have attempted to address all of the reviewer’s suggestions (attached).

Reviewer 3 Report

The revision has addressed my concerns. Thanks.

Author Response

In the previous submission, we have thoroughly addressed all the concerns raised by the reviewers. As of this round, we have no additional information or updates to provide.